# Sustainability of Impacts of Poplar Growth on Soil Organic Matter in Eutric Cambisols

**Christel Baum [1]**, **Martin Barth [1]**, **Kathrin Henkel [2]**, **Meike Siebers [3,†]**, **Kai-Uwe Eckhardt [1]**, **Ines Hilke [2]**, **Gerd Gleixner [2]** and **Peter Leinweber [1,*]**

[1] Soil Science, University of Rostock, Justus-von-Liebig-Weg 6, D-18051 Rostock, Germany; christel.baum@uni-rostock.de (C.B.); kultur@rg-gruenau.de (M.B.); kai-uwe.eckhardt@uni-rostock.de (K.-U.E.)

[2] Max-Planck-Institute for Biogeochemistry, Hans-Knöll Str. 10, D-07701 Jena, Germany; khenkel@bgc-jena.mpg.de (K.H.); ihilke@bgc-jena.mpg.de (I.H.); gerd.gleixner@bgc-jena.mpg.de (G.G.)

[3] Institute of Molecular Physiology and Biotechnology of Plants, University of Bonn, Karlrobert-Kreiten-Str. 13, D-53115 Bonn, Germany; meike.siebers@hhu.de

[*] Correspondence: peter.leinweber@uni-rostock.de; Tel.: +49-381-498-3100; Fax: +49-381-498-3122

[†] Present address: Max-Planck-Institute for Plant Breeding Research, Carl-von-Linné-Weg 10, 50829 Cologne, Germany.

**Abstract:** Short rotation coppices (SRC) with poplar on arable soils constitute no-till management in combination with a changed litter quality compared to annual crops. Both tillage and litter quality impact soil organic matter (SOM) composition, but little is known on the sustainability of this impact at the molecular level. We compared the microbial colonization and SOM quantity and quality of a young (4 years), old (17 years) and a former SRC with hybrid poplar (*Populus maximoviczii × Populus nigra* cv. Max) to adjacent arable sites with annual crops or grass. Total fungal and arbsucular mycorrhizal fungal phospholipid fatty acid (PLFA) markers were increased under no-till treatments with permanent crops (SRC and grass) compared to tilled cereals. Enrichments in fungal biomass coincided with C accumulation close to the soil surface (0–5 cm) but was abolished under former SRC after return to annual tillage. This management change altered the spatial distribution but not the accumulation of SOM within the topsoil (0–30 cm). However, lasting qualitative changes in SOM with increased proportions of lignin, lipids and sterols were found under current and former SRC. Increased colonization by arbuscular mycorrhizal fungi was correlated with increased invertase activity ($R = 0.64$; $p < 0.05$), carbohydrate consumption and a corresponding accumulation of lignins and lipids in the SOM. This link indicates a regulatory impact of mycorrhizal fungi on soil C dynamics by changing the quality of SOM. Increased stability of SOM to microbial degradation by higher portions of lipids and sterols in the SOM were assumed to be a sustainable effect of poplar growth at Eutric Cambisols.

**Keywords:** Pyrolysis-field ionisation mass spectrometry (Py-FIMS); Populus; PLFA; phospholipids; SOM quality

---

## 1. Introduction

Perennial woody biomass crops are of increasing economic relevance [1], leading to an expansion of poplars (*Populus* spp.) in short rotation coppice (SRC) on arable soil. Besides biomass production, C sequestration is a fundamental ecological aim of diverse agroforestry systems, including SRC [2,3]. SRC on arable soils leads to various soil ecological impacts such as changes in the microbial community and an increased litter quantity caused by winter harvest in the leafless state and woody litter with a

higher C/N ratio [4]. The elevated litter input from poplars in combination with the no-till management in SRC increased the C storage in soil organic matter (SOM) in former arable soils in Northern Germany [5]. Similarly, hybrid poplars grown for more than 4 years created a net C sink in Northern American agricultural soils [6]. However, there are also reports on initial C losses under SRC [7] and increased portions of less-stable C fractions under poplar, which suggest an accelerated organic matter decomposition in the following land use [8]. After transformation of former SRC to tilled arable use, the soil aggregate stability decreased relatively fast [5], but the chemical and biological reasons are not known in detail. In general, the effects of poplars in SRC on the soil C stocks differ site-specifically [9]. The sustainability of SOM, which was formed under SRC at different Cambisols is hardly known so far.

Generally, changes in SOM stocks may originate from changes in the C concentrations of soil and/or changes in the soil bulk density or both. As SOM consists of many different compound classes that differ in their resistance against microbial decomposition, any changes in C concentrations are almost always linked with alterations in the molecular composition of the SOM. Poplar in SRC may alter the SOM composition directly through a plant- or clone-specific litter input [10] and indirectly by an altered microbial diversity [4]. Moreover, mycorrhizal fungi may alter the litter decomposition significantly, contribute to the soil enzyme activities [11] and increase the fungal-to-bacterial ratio [12]. The external mycelium of the mycorrhizal fungi can be the dominant pathway through which C enters SOM under poplar in SRC [13]. The mycorrhizal mycelium in soil can amount to up to 900 kg ha$^{-1}$ [14], accounting for about 2.2% of the SOM in total and about half of the biomass of soil organisms. Thus, not surprisingly, poplar clones had a significant site-specific impact on the SOM composition and soil microbial communities in Luvisols from Saskatchewan [15]. In other Canadian soils, only modest differences in surface SOM chemistry related to land use were revealed by C and N *K*-edge X-ray absorption near edge fine structure spectroscopy (XANES) in conjunction with pyrolysis-field ionization mass spectrometry (Py-FIMS) [16]. However, the specific impact of fungal colonization on the SOM accumulation has not been elucidated so far, although fungal necromass was recently suggested to be a main microbial source to stable SOM [17].

Phospholipid fatty acid (PLFA) analyses can be used to describe the structure of microbial communities, as some PLFAs are indicators of specific microbial taxonomic groups [18]. The fatty acid 18:2ω6,9 is used as a fungal marker, including saprotrophic and ectomycorrhizal fungi, but it also may originate from cyanobacteria, algae and protozoa [19]. The fatty acid 16:1ω5 is commonly used as a marker of AM fungi [20]. The sum of PLFA, considered to be predominantly of bacterial origin (i15:0, a15:0, 15:0, i16:0, 16:1ω9, 16:1ω7t, i17:0, a17:0, 17:0, cy17:0, 18:1ω7, and cy19:0), was used as an index of bacterial biomass [21]. Therefore, in addition to the above Py-FIMS [15,16], the determination of PLFAs appears useful to assess qualitative differences in the microbial composition of SOM.

Furthermore, the phospholipid composition can disclose differences in the soil microbial community structure because the phospholipid composition differs between the various taxa (bacteria and fungi). Phospholipids, being essential membrane components, are also correlated with the living soil microbiota [22]. Thus, PLFA analysis can be used to obtain more detailed information to assess management-induced changes in the structure of the soil microbial community.

The objectives of the present study were to investigate (1) how poplars in SRC affected the quantity and quality of SOM of arable topsoil, and (2) the legacy SOM patterns from SRC in the subsequently tilled soils. We hypothesize that the introduction of a perennial wooden crop like poplar can sustainably promote C sequestration in arable soils by enhancing the stability of the SOM caused by woody litter and ectomycorrhizal symbiosis (the 'Gadgil-effect').

## 2. Materials and Methods

### 2.1. Study Sites and Soil Sampling

The three study sites (Cahnsdorf (CAH), Gülzow (GUL), Vipperow (VIP)) are located in northeastern Germany (Table 1). At each site, two test variants were investigated: (I) SRC with

hybrid poplar clones either at present or in the past and (II) arable reference (REF). The REFs at each site were selected to have continuous arable use without poplar or any other wooden crop for the previous decades and similar general soil properties as the SRC. REFs were selected within the shortest possible distance to the SRC (CAH, GUL) or former SRC (VIP) and were presumed to reveal the site-specific basic conditions under arable use in long-term absence of woody and ectomycorrhizal crops.

The reference variants—crop rotations—were dominated by cereals or fodder grasses. Winter wheat predominated in crop rotations at the reference sites and the former SRC fields. The dominating major soil units at all three sites soils are Eutric Cambisols according to the world reference basis (WRB) classification [23].

At the Cahnsdorf (CAH) study site, the annual means of temperature and precipitation are 9.6 °C and 548 mm, respectively [24]. The soil texture is 75% sand, 23% silt and 2% clay. The soil $pH_{CaCl2}$ was about 6.2 without significant differences between the treatments. The plot size was 4.50 m × 4.20 m. Five replicates per clone were established in a randomized block design with 33 poplar and willow clones in total. The planting was done in double rows with 1.50 m distance between two double rows and 0.75 m distance within the double row. The distance between the cuttings in the row was 0.60 m. SRC at this site has been established for four years (Table 1), thus we call it a relatively young SRC. The understory vegetation in this young SRC was still dense and dominated by grasses.

The study site Gülzow (GUL) is characterized by annual means of 8.2 °C temperature and 559 mm precipitation. The soil texture of the topsoil (0–30 cm) is 71% sand, 24% silt and 5% clay. The soil $pH_{CaCl2}$ was about 6.8 without significant differences between the treatments. The experimental SRC, established on a former arable soil in spring 1993, involved willows and poplar clones (n = 28) in a randomized block design with three replicates per treatment and a total size of 1.4 hectares. Each plot has a total size of 135 m$^2$ (three rows of 30 m length) [25]. The plant spacing was 1.50 m × 0.50 m. This means a total of 13,330 cuttings per hectare. No fertilizer or pesticides were applied so far. The tested treatment was harvested in January or February of 1996, 1999, 2002, 2005, 2008, 2011 (3-year-rotation). This SRC has been established for 17 years and can be considered as an old SRC. The understory vegetation of this old SRC is sparse, due to the shading, and comprised few grasses and herbs.

The study site Vipperow (VIP) is characterized by an average annual temperature of 8.0 °C and an average annual precipitation of 640 mm. The soil texture is 73% sand, 22% silt and 5% clay. The soil $pH_{CaCl2}$ was about 5.0 without significant differences between the treatments. A SRC with willows and poplar clones (24 in total) was established at this site on former arable soil in spring 1993. The plant spacing and harvesting was identical to the GUL test site. However, in 2007, after 14 years of use as SRC, it was finally harvested and returned to annual arable crops. The return was managed with a rotary hoe and a field cultivator down to 40 cm and 30 cm soil depth, respectively. The arable reference site of VIP was grown with grassland (*Lolium perenne*) from 2009–2010, i.e., two years of no till management before the sampling. Here we can investigate the lasting effects of a former SRC.

Soil samples were taken in April 2011. In the present SRC, any litter layer was removed before soil sampling. The SRC samplings in CAH were done at the end of the first rotation period and at GUL directly after harvesting the shoots of the 6$^{th}$ rotation period. Soil cores were taken from the upper topsoil (0–5 cm) for analyses of the mass spectrometric molecular chemical composition, PLFA concentrations and soil enzyme activities. This part of the topsoil was selected for supposedly being the most strongly affected by fungal colonization and the crop specific impact on the SOM. Additionally five independent cores of 8 cm diameter were taken per replicate in the topsoil, yielding 15 cores per treatment for the determination of the bulk density and C stock. The cores were divided into 5 depth increments: 0–5 cm, 5–10 cm, 10–15 cm, 15–20 cm and 20–30 cm.

## 2.2. Determination of the Soil Carbon (C) Contents and C Stocks

Samples were air-dried (except for analyses of PLFA concentrations and soil enzyme activities), screened for fine roots and stones, sieved at 2 mm, homogenized by grinding, and weighted prior to

quantifying the total carbon ($C_t$) and total nitrogen ($N_t$) contents by elemental analyses (varioMax, Elementar Analysensysteme GmbH Hanau, Germany). Organic carbon ($C_{org}$) was determined indirectly by $C_{org} = C_t - C_{inorg}$. Inorganic carbon ($C_{inorg}$) was measured after thermal pretreatment (muffle furnace, Heraeus, 450 °C, 16 h) of a subsample by elemental analysis as well [26].

The calculation of carbon stocks (*C-stock_i*) follows Equation (1), where the $C_{org}$ in the sampled soil depths (*i*) was related to the layer thickness (*TH_i*), the content of stones (*ST_i*), and the bulk density (*BD_i*) in the corresponding soil segments. *BD_i*, which is the weight of dry soil (*m_i*) divided by the total soil volume (*V_i*) in each layer, was calculated according to Equation (2).

$$C-stock_i\left[\frac{kg}{m^2}\right] = Corg_i\left[\frac{g}{kg}\right] * BD_i\left(1 - \frac{ST_i[\%]}{100\%}\right)\left[\frac{g}{cm^3}\right] * TH_i[cm] * 0.01 \qquad (1)$$

$$BD_i\left[\frac{g}{cm^3}\right] = \frac{m_i[g]}{V_i[cm^3]} \qquad (2)$$

### 2.3. Molecular-Chemical Composition of the Soil Organic Matter

The molecular–chemical composition of the SOM was determined by pyrolysis field-ionization mass spectrometry (Py-FIMS) according to Schulten and Leinweber [27]. For each analysis, 0.5 mg of finely ground soil were thermally degraded using a modified Finnigan MAT 95 high-performance mass spectrometer. The samples were heated in a high vacuum from 50–750 °C at a heating rate of 10 K per magnetic scan. During 20 min of total registration time, about 60 magnetic scans were recorded for the mass range of 15–900 Da. These were combined to obtain one thermogram of total ion intensity (TII) and an averaged Py-FI mass spectrum. The summed spectra of at least three replicates were averaged to give the final survey spectrum. These survey spectra, in particular the assignment of marker signals to ten important classes of chemical compounds, were interpreted as described by Leinweber et al. [28]. These compound classes are carbohydrates with pentose and hexose subunits, phenols and lignin monomers, lignin dimers, lipids, alkanes, alkenes, bound fatty acids and alkylmonoesters, alkylaromatics, mainly heterocyclic N-containing compounds, sterols, peptides, suberin and free fatty acids. A series of marker signals and the volatilization temperature were considered for identification. For each of the 60 single scans, the ion intensities of these marker signals were calculated. The average ion intensities for each class of compound were plotted against the pyrolysis temperature, giving characteristic thermograms. All samples were weighed before and after Py-FIMS to normalize the ion intensities per mg sample.

### 2.4. Isolation of Total Lipids from Soil and Quantification by Quadrupole Time-of-Flight Mass Spectrometry

Lipids for mass spectrometry were extracted according to Kruse et al. [29]. Briefly, soil samples (fine earth < 2 mm, ~5 g wet weight) were placed into a glass tube with a Teflon lined screw cap. A first extraction was done with 10 mL of chloroform/methanol/formic acid (1:1:0.1 *v/v*) and the organic phase collected. The solvent of the lipid extract was evaporated under a stream of $N_2$. The lipid extraction was repeated with 8 mL CHCl$_3$/MeOH (2:1 *v/v*) and the organic phases were combined. Two milliliters of aqueous 1 M KCl/0.2 M H$_3$PO$_4$ was added to the combined chloroform extracts. Samples were vortexed and centrifuged (2000× *g*, 5 min). The organic phase was harvested. The remaining soil sample was dried at 105 °C for 48 h and used to determine the dry weight. The solvent of the lipid extract was evaporated under a stream of $N_2$. Total lipids were dissolved in 1 mL of chloroform. Due to technical problems, two treatments (CAH-REF, VIP-REF) had to be excluded from the analyses and only four treatments can be presented in Figure 4.

Lipids were quantified using a Q-TOF (quadrupole time-of-flight) mass spectrometer (Q-TOF 6530; Agilent Technologies) according to Gasulla et al. [30]. For quantification, a phospholipid standard mix containing two molecular species each of phosphatidylserine (PS), phosphatidylcholine (PC), phosphatidylethanolamine (PE), phosphatidylglycerol (PG), phosphatidylinositol (PI) and phosphatidic acid (PA) was added to the phospholipid extract obtained from the soil. The phospholipid extract

was infused into the Q-TOF mass spectrometer using the Agilent ChipCube nanospray source. Phospholipids were quantified by MS/MS experiments in the positive mode [31].

## 2.5. PLFA Analyses

PLFAs were extracted according to the method of Bligh and Dyer [32] and Zelles and Bai [33] from 70 g of fresh soil per sample. Soil lipids were extracted using a mixture of chloroform, methanol and phosphate buffer (1:2:0.8 *v/v/v*). Phospholipids were isolated on silica columns and transmethylated using a methanolic KOH solution. Saturated, polyunsaturated and monounsaturated fatty acid methyl esters (FAME) were separated into fatty acids using aminopropyl-modified and silver-impregnated SPE columns (Varian). The samples were quantified with a GC-AED System (GC: HP 6890 Series, AED: G 2350 A, Agilent Technologies, United States) using a BPX 70 column (50 m × 0.32 mm I.D., 0.25 μm film thickness) in the split mode (10:1). Helium was used as a carrier gas at a flow rate of 1.3 mL min$^{-1}$. The temperature program started at 100 °C (for 1 min). Thereafter, the temperature was raised to 135 °C at a rate of 4 °C min$^{-1}$, then to 230 °C at 2 °C min$^{-1}$. The final temperature of 260 °C was reached after further raising the temperature at 30 °C min$^{-1}$, and was kept constant for 1 min. PLFAs were identified by a comparison with a standard mixture (Supelco) of saturated fatty acids and unsaturated fatty acids and by using mass spectral data reported by Gattinger et al. [34]. The amount of phospholipid fatty acid (PLFA) 18:2ω6,9 was used as an indicator for fungal biomass in general [35] and 16:1ω5 as an indicator for the biomass of AMF [20].

## 2.6. Soil Enzyme Analyses

Soil enzyme activities were measured in sieved (<2 mm) fresh soil samples by determining glucose released from added substrates i.e., xylan (for xylanase) and sucrose (for invertase). This was done by incubating soil samples with the substrate solution for 3 h (invertase) or 24 h (xylanase) at 50 °C and measuring the release of glucose equivalents (GE) using a photometric method described and validated by Schinner and von Mersi [36].

## 2.7. Statistical Analyses

Two-factorial analyses of variance (2-factorial ANOVAs) were calculated to prove differences in the C-stock, molecular chemical composition and soil enzyme activities caused by the factors management, sampling date, site and their interactions and of management separately for each site. Data were checked for normality and homogeneity of variance and transformed if necessary. Multivariate detrended correspondence analysis (DCA) was elaborated for two distinct matrixes: (i) all PLFA markers and (ii) the chemical composition of the SOM (compound class from Py-FIMS analysis). Additionally, markers of microbial groups were analyzed by simple correlation analysis. Statistical analyses were performed using SPSS version 20 and R version 2.13.1.

## 3. Results

### 3.1. Soil Organic Matter Quantity

The C$_{org}$ and N$_t$ concentrations in the upper 0–5 cm of soil under current SRCs at CAH and GUL exceeded those of the REF (Table 1). The soil at the former SRC at VIP had lower C$_{org}$ and N$_t$ concentrations than the REF site. The C/N ratio was higher in soil under long-term SRC at site GUL, not affected by the treatment at site CAH and slightly lower under former SRC at VIP compared to the corresponding REF.

**Table 1.** Site coordinates, treatment, period of use as SRC, $C_{org}$-, $N_t$- content and C-to-N ratio in the soil depth of 0–5 cm at the test sites Cahnsdorf (CAH), Gülzow (GUL) and Vipperow (VIP) in Germany.

| Site with Abbreviation | Coordinates | Treatment | Use as SRC, Poplar Clone | $C_{org}$ (mg g$^{-1}$) | $N_t$ (mg g$^{-1}$) | C-to-N |
|---|---|---|---|---|---|---|
| Cahnsdorf (CAH) | 51°51′ N, 13°45′ E | SRC | since 2006 (4 years) Max | 12.6b (±1.5) | 1.2b (±0.1) | 10.9a (±0.5) |
| | | REF | | 8.3a (±0.6) | 0.9a (±0.1) | 9.9a (±0.5) |
| Gülzow (GUL) | 53°48′ N, 12°3′ E | SRC | since 1993, (17 years) Max | 15.6b (±2.3) | 1.3b (±0.2) | 12.6b (±0.8) |
| | | REF | | 7.1a (±0.5) | 0.8a (±0.1) | 9.4a (±0.6) |
| Vipperow (VIP) | 53°19′ N, 12°41′ E | fSRC | 1993–2007, (14 years) Max | 8.1a (±0.8) | 0.9a (±0.1) | 9.0a (±0.7) |
| | | REF | | 15.6b (±3.8) | 1.5b (±0.3) | 10.7b (±0.5) |

SRC—short rotation coppice with poplar, REF—tilled arable site, fSRC—former short rotation site now in tilled arable use. Different letters within the site indicated significant differences between the variants of treatment ($p < 0.05$).

The bulk density at 0–30 cm soil depth was about 1.3 g cm$^{-3}$; there was no significant difference between the SRC and arable REF. Under both present SRC treatments, an increased C-stock close to the surface (0–5 cm soil depth) was observed (Figure 1). At the tilled REF sites (Cahnsdorf and Gülzow), the highest C-stock was measured in 15–20 cm soil depth. The total C-stock in 0–30 cm unexpectedly did not differ significantly between the treatments at the short-term SRCs CAH and at the former SRC VIP in comparison to the adjacent REF treatments. The C-stock was significantly affected by the site, the management and the interaction of site x management in each tested soil depth (Table 2).

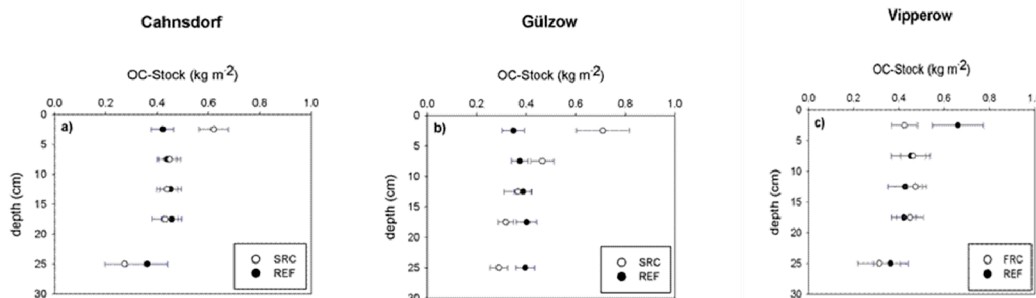

**Figure 1.** OC-stock in the topsoil (0–30 cm soil depth) of (**a**) 4-year old (Cahnsdorf); (**b**) 15-year old (Gülzow) and (**c**) former (Vipperow) short rotation coppice (SRC), former SRC (FRC) and adjacent reference sites with cereals or manure grasses (REF).

**Table 2.** Test values of the F distribution from the analyses of variance (ANOVA) of the C-stocks from three study sites in Germany (poplar in short rotation coppice vs. annual arable crops).

| Factor | Soil Depth | | | | |
|---|---|---|---|---|---|
| | 0–5 cm | 5–10 cm | 10–15 cm | 15–20 cm | 20–30 cm |
| Intercept | 5006.879 | 8844.388 | 8885.768 | 8654.424 | 3760.469 |
| Site | 176.012 ** | 459.629 ** | 337.368 ** | 261.747 ** | 132.831 ** |
| Management | 65.876 ** | 53.423 ** | 56.577 ** | 67.079 ** | 72.177 ** |
| Site × management | 7.904 ** | 91.025 ** | 54.491 ** | 32.147 ** | 51.359 ** |

** $p < 0.001$; * $p < 0.005$.

### 3.2. PLFA Pattern and Soil Enzyme Activities

The content of the PLFA 18:2ω6,9, as a marker of the total fungal colonization, ranged from 11.3 to 17.7 ng g$^{-1}$ with lowest content under former SRC at site VIP and highest content under short-term SRC at site CAH. It differed in a treatment-specific manner at each site with higher contents under present SRC than under REF at site CAH and GUL and lower contents under former SRC than under REF at site VIP.

The contents of the PLFA 16:1ω5, indicating colonization with AMF, although containing bacterial contributions, ranged from 4.1 to 13.1 ng g$^{-1}$. In this range, the former SRC at site VIP had the lowest and the short-term SRC at site CAH the highest content of the PLFA 16:1ω5. The total bacterial PLFA content was lowest under REF at site CAH and highest under SRC at this site. The ratio of bacterial to fungal PLFA contents was significantly higher under SRC than under REF at CAH and GUL, but the difference was not significant under former SRC and REF at site VIP.

The invertase activity in soil ranged from 310 to 829 mg GE g$^{-1}$ 3 h$^{-1}$ (Table 3) and differed in a site-specific manner with lowest activity at VIP and highest activity at CAH. At each site, treatment-specific differences were revealed. Soil at the site CAH revealed significantly higher invertase activities under SRC than under REF. The invertase activity was significantly correlated with the contents of the PLFA 16:1ω5 ($R^2 = 0.46$, $p < 0.05$, n = 18, Figure 2).

The xylanase activity ranged from 295 to 885 mg GE g$^{-1}$ 24 h$^{-1}$ with lowest activity under SRC at site GUL and highest activity under SRC at site CAH. Soil at the sites GUL and VIP revealed

significantly lower xylanase activity under present or former SRC than under REF. The site-specific effects and the interaction of management and site effects on soil enzyme activities were significant.

**Table 3.** PLFA contents, ratio of bacterial to fungal PLFAs, invertase and xylanase activities of the topsoil of three study sites in Germany under two different management systems (poplar in SRC or former SRC, fSRC vs. annual arable crops as REF) (standard deviation in parentheses, n = 3).

| Site | Treatment | PLFA 18:2ω6,9 (ng g$^{-1}$) | PLFA 16:1ω5 (ng g$^{-1}$) | PLFA Total Bacterial (ng g$^{-1}$) | Ratio Bacterial/ Fungal PLFAs | Invertase (mg GE g$^{-1}$ 3h$^{-1}$) | Xylanase (mg GE g$^{-1}$ 24 h$^{-1}$) |
|---|---|---|---|---|---|---|---|
| Cahnsdorf | SRC | 17.7b (±0.4) | 13.1b (±1.3) | 178.8b (±25.4) | 10.1b (±1.6) | 829b (±79) | 885b (±30) |
|  | REF | 12.1a (±4.2) | 5.0a (±1.9) | 78.5a (±23.3) | 6.8a (±0.9) | 535a (±41) | 796a (±64) |
| Gülzow | SRC | 15.3b (±1.3) | 12.7b (±2.1) | 157.6b (±23.5) | 10.2b (±0.6) | 560a (±155) | 295a (±41) |
|  | REF | 13.1a (±1.4) | 6.4a (±0.9) | 98.2a (±17.1) | 7.4a (±0.7) | 699b (±94) | 631b (±55) |
| Vipperow | fSRC | 11.3a (±1.6) | 4.1a (±0.6) | 98.6a (±8.8) | 8.8a (±1.1) | 310a (±11) | 491a (±55) |
|  | REF | 17.1b (±2.3) | 7.0b (±1.9) | 152.2b (±25.5) | 8.9a (±0.1) | 360b (±26) | 557b (±105) |

SRC—short rotation coppice with poplar, REF—tilled arable site, fSRC—former short rotation site now in tilled arable use, GE—glucose equivalents. Different letters within the site indicate significant differences between the variants of treatment ($p < 0.05$).

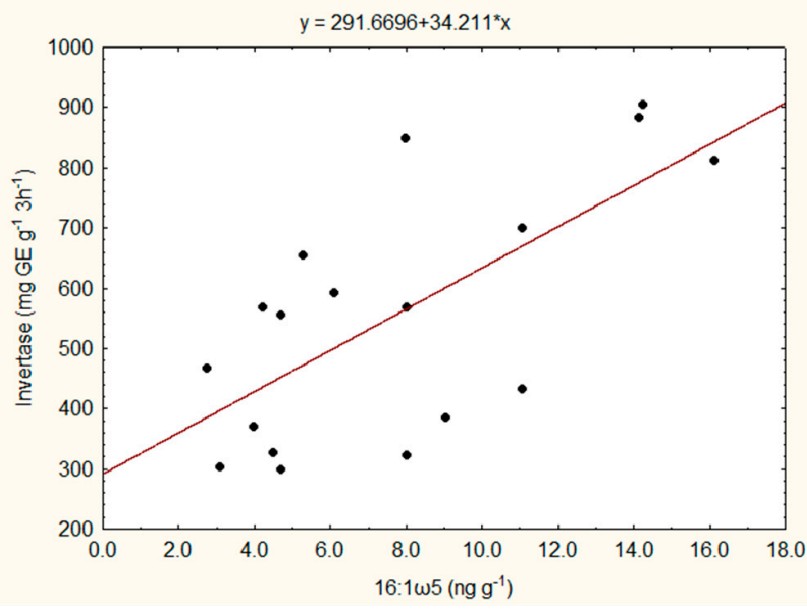

**Figure 2.** Linear regression between the amount of PLFA 16:1ω5 and the invertase activity in the soil (0–5 cm depth) of test sites with two managements: under short rotation coppice (SRC) or former short rotation coppice (fSRC) and reference sites with cereals or grassland (REF) (three test sites with n = 6 test plots each; 18 plots in total ($R^2 = 0.46$, $p < 0.05$, n = 18).

### 3.3. SOM Composition

The Py-FI mass spectra of the soil differed strictly in a site-specific manner (see Figure 3 for sites GUL and VIP). Generally, the Py-FI mass spectra were dominated by signals of carbohydrates (*m/z* 60, 96, 114, 126) and phenols and lignin monomers (*m/z* 124, 150, 194, 208). Signals of lipids (*m/z* 256, 392) were generally more intensive at site VIP under both treatments than at site GUL (*m/z* 256).

Furthermore, the signal of a flavonoid (*m/z* 308) was characteristic for test site VIP in both treatments. The thermograms of total ion intensity (TII) (Figure 3, inserts upper right and Table 4) clearly showed larger ion intensities at site VIP than at site GUL and, at the same site, larger TIIs under SRC and former SRC than under the adjacent REF.

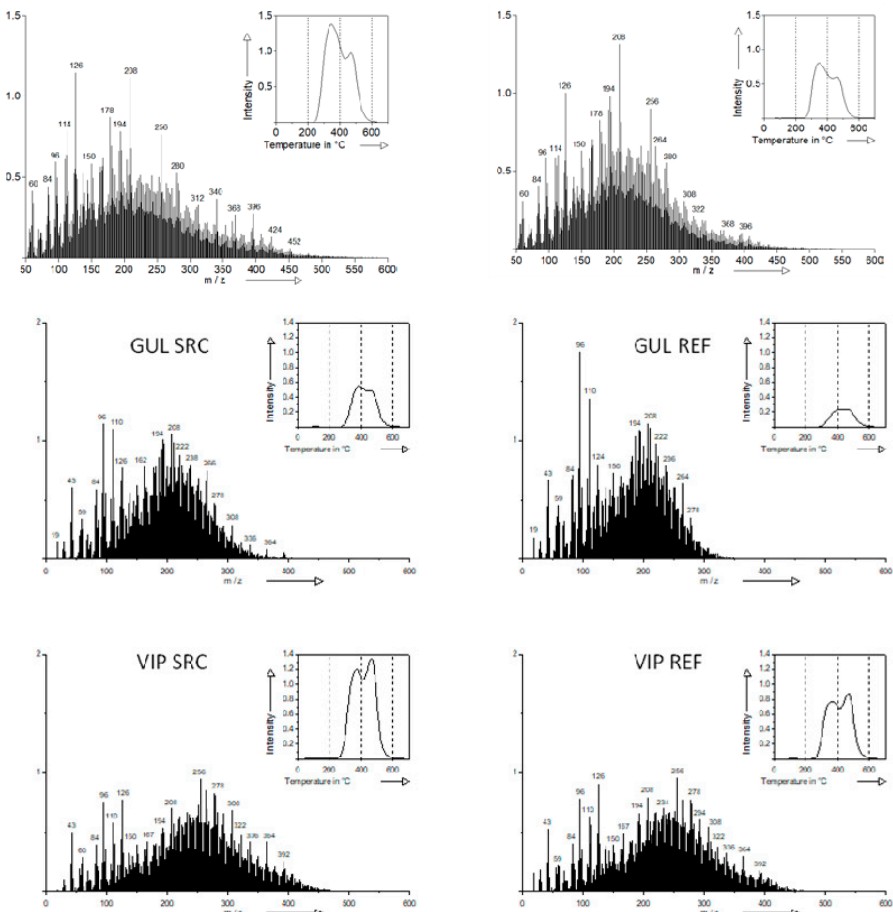

**Figure 3.** Py-FI mass spectra and thermograms (inserts) of the upper topsoil (0–5 cm soil depth) under a 15-year old short rotation coppice (SRC) at the test site Gülzow and a former SRC with hybrid poplar clones 4 years after return to tilled annual crops and with continuously tilled annual crops at the test site Vipperow and adjacent reference sites with cereals or grasses (REF) at both sites.

Differences between treatments were reflected similarly by the proportions of compound classes from Py-FIMS analyses, total lipid extracts and the single PLFA pattern (see Tables 3 and 4). The relative abundance of carbohydrates in the SOM was lower under SRC or former SRC than under the REF at all sites (Table 4). The relative abundances of phenols and lignin monomers also were lower under long-term SRC (GUL) and former SRC (VIP) but the difference was not significant under short-term SRC and the adjacent REF at site CAH. Lignin dimers had larger proportions in the SRC at sites GUL and VIP. Accordingly, at these sites the ratio of phenols/lignin monomers to lignin dimers was smaller compared to the reference samples. The relative abundance of lipids and sterols was larger under SRC than under the REF at all three sites. The relative abundance of alkylaromatics was lower under SRC than under REF but did not differ significantly between former SRC and the adjacent REF at site VIP. The relative abundance of peptides differed significantly between the treatments only at site GUL where it was larger under the REF than under SRC. At site VIP the relative abundance of free *n*-$C_{16}$ to *n*-$C_{34}$ fatty acids was insignificantly smaller under former SRC than under the adjacent REF.

**Table 4.** Total ion intensity (TII) and relative abundance of 10 compound classes of soil organic matter under short rotation coppice (SRC), former SRC (fSRC) and reference sites with cereals or grass (REF) at three test sites (Cahnsdorf, Gülzow, Vipperow) in Germany (means of 3 replicates, ±SD given in brackets).

| Site Andvariant | TII | Carbohydrates | Phenols, Ligninmonomers | Lignin Dimers | Lipids, Alkanes, Alkenes, Fatty Acids, n-alkyl Esters | Alkylaromatics | N-containingcompounds | Sterols | Peptides | Suberins | Fatty Acids n-C16 to n-C34 |
|---|---|---|---|---|---|---|---|---|---|---|---|
| Cahnsdorf SRC | 29.7b (±5.0) | 4.4a (±0.1) | 8.8a (±0.3) | 4.2a (±0.3) | 10.1b (±0.6) | 10.6a (±0.5) | 6.8a (±0.3) | 2.1b (±0.2) | 2.0a (±0.1) | 0.2b (±0.0) | 2.7b (±0.2) |
| REF | 20.8a (±0.9) | 4.6b (±0.0) | 8.8a (±0.1) | 5.0b (±0.1) | 9.6a (±0.0) | 11.2b (±0.1) | 6.7a (±0.1) | 1.6a (±0.0) | 1.9a (±0.0) | 0.1a (±0.0) | 2.1a (±0.1) |
| Gülzow SRC | 11.4b (±3.1) | 6.8a (±0.1) | 13.2a (±0.3) | 3.9b (±0.2) | 7.1b (±0.3) | 12.2a (±0.3) | 7.9a (±0.2) | 1.0b (±0.1) | 2.6a (±0.1) | 0.0a (±0.0) | 0.9b (±0.1) |
| REF | 5.0a (±0.9) | 7.8b (±0.0) | 15.2b (±0.1) | 3.0a (±0.1) | 5.2a (±0.1) | 13.2b (±0.1) | 8.8b (±0.1) | 0.0a (±0.0) | 3.0b (±0.0) | 0.0a (±0.0) | 0.2a (±0.0) |
| Vipperow fSRC | 22.0b (±1.5) | 5.2a (±0.1) | 8.7a (±0.0) | 5.2b (±0.1) | 9.7b (±0.2) | 10.9a (±0.1) | 6.8a (±0.1) | 2.0b (±0.1) | 2.1a (±0.0) | 0.1a (±0.0) | 2.1a (±0.1) |
| REF | 17.1a (±1.5) | 5.7b (±0.1) | 8.9b (±0.1) | 3.4a (±0.0) | 9.3a (±0.1) | 10.8a (±0.1) | 7.2b (±0.1) | 1.6a (±0.0) | 2.2a (±0.1) | 0.1a (±0.0) | 2.2a (±0.0) |

Different letters within the site indicate significant differences between treatments ($p < 0.05$).

The site-specific phospholipid content under SRC and former SRC decreased in the following order: GUL-SRC (26.9 nmol g$^{-1}$) > CAH-SRC (15.5 nmol g$^{-1}$) > VIP-fSRC (7.6 nmol g$^{-1}$) and was generally lowest in the arable reference GUL (GUL-REF, 5.9 nmol g$^{-1}$). PC, PE and PI were the most abundant classes of phospholipids at all tested sites and variants. PC mainly occurs in eukaryotic cells (fungi, nematodes, animals) but it also can be found in certain bacteria to a lesser extent [37]. The major bacterial phospholipid classes were PE (constituting 50 to 90% of the phospholipids), and PG [38]. Other phospholipids can be found in minor amounts in bacteria. PE is also found in fungi, next to PC and PI, which represent the main phospholipid classes.

The total content of phospholipids (5.9 to 26.9 nmol g$^{-1}$) in the soil differed more between SRC and REF at the same site (GUL) than between different sites (CAH and VIP; see Figure 4).

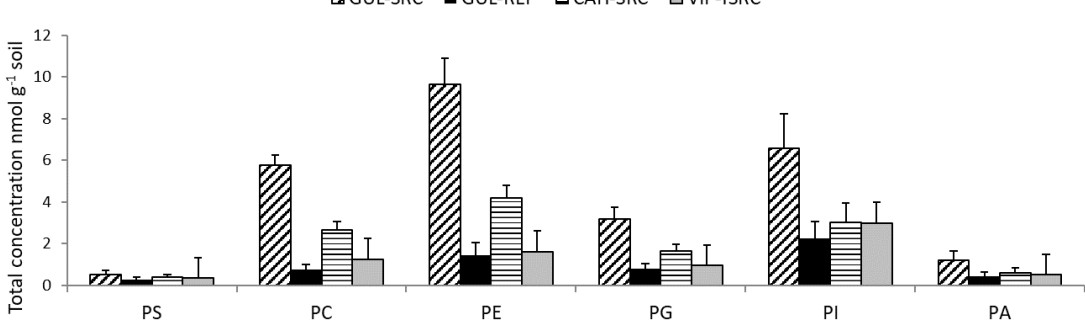

**Figure 4.** Phospholipid composition of the upper topsoil (0–5 cm soil depth) under short rotation coppice (SRC) and a reference site with cereals (REF) at the test site Gülzow (GUL) and Cahnsdorf (CAH) and a former SRC (fSRC) with hybrid poplar clones 4 years after return to tilled annual crops at the test site Vipperow (VIP) as determined by Q-TOF MS/MS analysis (n = 5): Total concentration of major phospholipid classes (phosphatidylserine, PS; phosphatidylcholine, PC; phosphatidylethanolamine, PE; phosphatidylglycerol, PG; phosphatidylinositol, PI; phosphatidic acid, PA) in nmol g$^{-1}$ soil dry weight.

## 4. Discussion

The near-surface C$_{org}$ accumulation, observed at 0–5 cm under current (CAH, GUL) but not under former SRC (VIP) (Figure 1), most likely originates from larger leaf and partly root litter inputs in combination with the no-till management [5,39] compared to the rather uniform C-stocks in the 5-cm-sections of the tilled reference samples. The effect of no-till management can be explained by the reduced soil disturbance, which generally maintains aggregation and, thus, prevents aggregate-protected C$_{org}$ from decomposition [40]. The near-surface C$_{org}$ accumulation under poplar agrees with De Gryze et al. [41], who reported increased profile C-stocks in a 10-year poplar plantation even to the level of native forests. In the present study, the C$_{org}$ accumulation under SRC got reduced after return to annual crops with tillage so that the SRC-induced increase in the topsoil C$_{org}$-stock was not sustained (site VIP; Figure 1). The higher C$_{org}$-stock in the upper 0–5 cm in the REF of the site VIP compared to the former SRC at this site is explained by the present grassland vegetation at REF, which stores more C$_{org}$ than the intensively tilled wheat cropping at the former SRC. This confirms a possibly lower C$_{org}$ accumulation under SRC than under grass as was observed for willows in SRC vs. grass on a Vertic Cambisols by Jandl et al. [42]. Litter composition in 0–5 cm soil depth under grassland is formed mainly by dense fine root litter, whereas leaf litter with high portions of waxes is characteristic of SRC [42]. As fine roots and rhizodeposits contain flavonoids, this might explain their relative enrichment under grass and cereals at the site VIP (see Figure 3). The lower microbial biomass and enzyme activities in soil under the former SRC vs. REF with grass at site VIP confirm results of Waldrop and Firestone [43] who investigated the same parameters in soil cores from an oak canopy transferred to a grassland environment. Higher precipitation at site VIP than at sites GUL and CAH (see Section 2.1) might have accelerated the changes in microbial activity and SOM turnover after changes in the land use.

Enhanced fungal colonization indicated by an increased content of the fungal PLFA 18:2ω6,9 (Table 3) can be caused by the lower tillage intensity as recently described in a comparison of different tillage systems [44]. However, only present SRC (at sites CAH and GUL) can be assumed to contain significant portions of ectomycorrhizal fungi within the total fungal biomass in the soil, as all other sites lack host plants of ectomycorrhizal fungi. However, the vegetation at all test sites of the present study contain host plants of AM fungi. Unlike the increased total and AM fungal PLFA content, the bacterial/fungal ratio of PLFAs (Table 3) was SRC-affected only under present SRC at CAH and GUL, but not under former SRC at VIP. A further specific difference within the fungal colonization under SRC and grass or cereals was the lower ratio of total fungal PLFA (18:2ω6,9) to AM fungal PLFA (16:1ω5) under SRC (1.2–1.3 vs. 2.0–2.7). This indicates a selective promotion of AM fungi within the fungal community under SRC, which is combined with increased bacterial colonization too.

The quantity and composition of the SOM is linked with the fungal colonization. Significant smaller proportions of carbohydrates in the soil under present SRC compared to the adjacent REF (Table 4) agreed with increased total fungal colonization in these treatments (Table 3) and might be explained by the high C demand of fungi [45]. The C demand of ectomycorrhizal fungi is particularly large during their saprotrophic growth periods [46], and thus soil carbohydrates were probably utilized at disproportionally larger extents under SRC. The impact of ectomycorrhizal fungi on the saprotrophic activity as it can be assumed from the present data (Figure 1, Tables 3 and 4) agrees with the so-called 'Gadgil-effect' [47]. The correlation of AM colonization and invertase activity ($R = 0.64$; $p < 0.05$; see Figure 2) agreed with findings of Zhang et al. [48] in an incubation experiment and might be based on increased substrate availability for saprotrophic microorganisms in the presence of AM fungi.

Our findings support the 'Priming Effect' hypothesis that mycorrhizal fungi decompose soil C (in our case from carbohydrates), when allocation of plant photosynthate to mycorrhizal roots is high, such that plant C primes the growth and activity of the mycorrhizal fungi [45]. These were revealed by the correlation of the colonization density of mycorrhizal fungi and invertase activity (see Figure 2) and lower relative abundance of carbohydrates under tilled arable use than under current SRC at the sites CAH and GUL and under the grassland (REF) at site VIP (see Table 4). Furthermore, the plant partner of ectomycorrhizal fungi (poplar in the present study) can increase its expression of hexose importer genes at the plant/fungus interface and the hexose uptake capacity of plant roots in combination with an increase in photosynthesis [49].

The higher proportion of lignin dimers in the soil under long-term SRC and former SRC soils in the present study might be a consequence of partial degradation of large litter amounts as described by Kalbitz et al. [50] for dissolved organic matter. The increased proportion of both lignin dimers and lipids in the soil also agreed with results from soil under SRC with willows (*Salix viminalis*) at Swedish test sites [43]. Multivariate detrended correspondence analysis (DCA) separated the present SRC and the former SRC in its SOM quality from the arable REFs. The distance between REF and SRC was shortest for the site VIP with former SRC and largest for the site GUL with long-term and present SRC (Figure 5). Larger precipitation at VIP (see Section 2.1) might have accelerated the alignment of the two different types of land use. Furthermore, the present vegetation at VIP (grass at REF and cereals at fSRC) can be assumed to have very similar litter quality.'

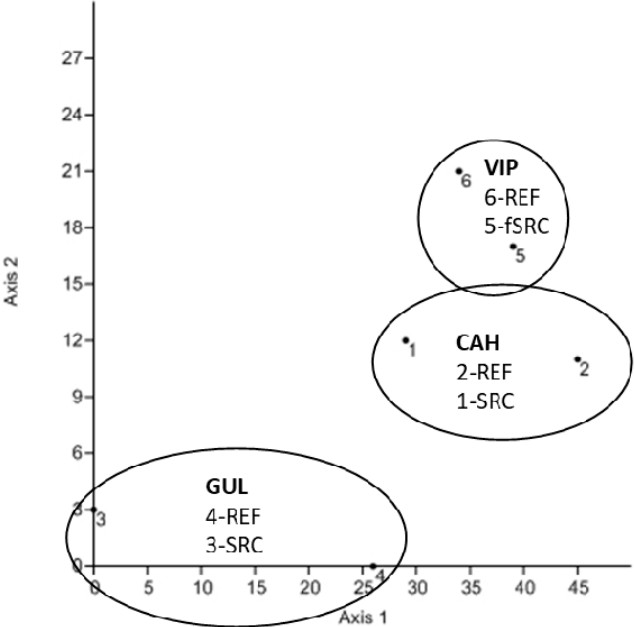

**Figure 5.** Multivariate detrended correspondence analysis (DCA) based on the relative abundances of substance classes in the soil organic matter (Py-FIMS) and the abundance of PLFA marker signals in the soil. Circles represent separate test sites with two managements: under short rotation coppice (SRC) or former short rotation coppice (fSRC) and reference sites with cereals or grassland (REF) (each point is based on three replicates per treatment).

Lignin is known to inhibit xylanase activity [51]. This might explain the decreased xylanase activities under long-term and former long-term SRC (see Table 3). Furthermore, decreased xylanase activity under SRC can be caused by a lower fine root density under poplar than under the cereals and grass, since a high root density is correlated with a high xylanase activity in the soil [52]. In agreement with the present results on C cycling, Kahle et al. [53] revealed decreased β-glucosidase activity in the topsoil under SRC with poplar at site GUL, which can also be inhibited by lignin [51].

The present study revealed the significance of woody plants on arable soils in SOM formation. The chance of increased SOM stability for increased C sequestration and decreased $CO_2$ release in the long-term should be the focus of subsequent analyses. Furthermore, options of use of ectomycorrhizal vegetation to support C sequestration in arable soils should be a focus, as it was confirmed to be site-specific. This supports the statement of Fernandez and Kennedy [47] for forest soils in advance.

## 5. Conclusions

The combined quantification of soil C-pools with analyses of the composition of SOM revealed treatment-specificity in the SOM quality and tillage-induced changes of the vertical distribution of SOM in the topsoil. Treatment effects can exceed the general site-specific differences in the topsoil of Eutric Cambisols.

Although termination of SRC and return to rotation with tillage resulted in a redistribution of SOM and no lasting C accumulation in the topsoil, the qualitative changes in SOM with increased proportions of lignin, lipids and sterols were lasting and might increase the SOM stability in the long-term.

Increased colonization by AM fungi was correlated with increased invertase activity and carbohydrate consumption and a corresponding lignin and lipid accumulation in the SOM. This link highlights a regulatory impact of mycorrhizal fungi in soil C dynamics by lasting changes in the SOM quality.

**Author Contributions:** Conceptualization, C.B., G.G. and P.L.; Data curation, M.B., K.H., I.H. and M.S.; Investigation, M.B., K.H., M.S., K.-U.E. and I.H.; Methodology, M.B., G.G., M.S. and P.D.; Supervision, C.B. and P.L.; Writing—original draft, C.B. and M.B.; Writing—review and editing, C.B., G.G. and P.L.

**Funding:** This work was financially supported by the German Research Foundation (DFG, BA 1494/3-1 and GL 262/13-1).

**Acknowledgments:** We thank A. Gurgel of the Institute of Crop production and Agricultural Economics Science, Research Centre for Agriculture and Fishery (Germany) for the management information on their test sites in Gülzow and Vipperow and D. Landgraf for the management information on the test site Cahnsdorf. The authors are thankful to P. Narf for very helpful suggestions concerning the lipid extraction method. We are grateful to the anonymous reviewers for their very valuable comments.

**Conflicts of Interest:** The authors declare no conflict of interest.

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
