# Peer review of "Sustainability of Impacts of Poplar Growth on Soil Organic Matter in Eutric Cambisols"

_soilsystems, doi:10.3390/soilsystems3020032_

Round 1
Reviewer 1 Report
The manuscript describes in detail analyses of SOM components in 3 experimental areas, with implications on accumulation, quality and stability of organic C.
The work is very systematic, the experimental design is well planned, and the field sites are meaningfully selected. However, the ms lacks novel hypotheses or questions that would address current knowledge gaps; the results serve as a mere confirmation of previously proposed, confirmed and published hypotheses on microbial / fungal role in C stabilization in soils.
The paper is very observational and descriptive, and the outcome is very predictable - that organic C and N has been, indeed, higher under SRCs. The main contribution of the study is the field setup, with a good general relevance to the SRC areas. The ms was somehow hard to read – even papers with a lot of comparison should have a good logical flow.
Construction of some sentences was rather cumbersome - I suggest a review of a technical writer.
Despite having said all this, I trust the topic of the revised manuscript will be of interest to readers of Soil systems.
I have general and specific comments for the authors’ consideration:
General:
· L 86 - as I said, the hypothesis is very broad. What is the specific novelty of this research? Point it out.
· Fig 2 and 3 - when comparing 6 sites, they need to be all included. Not clear how sites were selected to be shown in 2 and 3. Controls (REF) are an important part of the data presented. Specifically, Fig 2 – you compare GUL and VIP, where is the young CAH? It is listed in Table 4.
· The whole normalization against controls was not clear. Adjusting the values of the respective analyses to controls should be more prominent.
· Discussion was very descriptive of observed results: readers may appreciate your opinion on what could be projected, based on your results and literature, to e.g. elevated atmospheric CO2 concentration and other climate change factors. On production, turnover and allocation to ECM fungi. C sink.
Specific:
(Note: the line numbering changes throughout the ms: it starts again from line 1 at four different places: p 7, 9, 11, 14. So I will refer to page/line.)
Page 1/ line 15 – dynamics- more like composition
1/25 - AM fungi used for the first time, write it out (it’s on 1/19).
1/28 – increased
p2/45 – no hyphen in land use
48 – a cumbersome sentence
56 – their à the
57 - In a sentence constructed this way, it is not clear what is ‘62%’ referring to.
56 – ‘The external mycelium …’ sentence should be placed following its intro on l. 58 – ‘The mycorrhizal …’,
82 – ‘structure of’- is repeated.
86 – these hypotheses are very broad.
p3/128 – were taken
130 – by à for – streamline the sentence.
131 - Strongest à the most strongly
p5/223.4 – the ratio is not wider or closer. It’s higher or lower.
Results:
3.1. - Was the enzymatic activity of the three SRCs normalized against their REF?
Fig 3 – Phospholipids: again – 2 of them are missing: ref to CAH and VIP. If you’re comparing all 6 sites, they need to be plotted.
VIP- FRC: ‘former’ should be previously defined. Later, fSRC is used in Fig 5. Please clarify.
p14/20 - surface-near à near-surface
25 - surface-near à near-surface
28 – disappeared – according to Fig 1, it didn’t disappear, it got reduced.
29 - sustainably (?) - sustained?
37, 38 – this belongs to Results.
Fig 4 is not mentioned in Results– it just comes in with a very little description, and no site-specificity. Just ‘soil”? What soil?
p15/57-63: the generalized priming effect correlated from literature to your experimental sites is not convincing. This paragraph would need more factual substance based on your results.
p. 15 – fSRC – explain f (first time used ‘former SRC’), also unexplained in Fig 3 VIP-FRC
Fig 5 – DCA has not been defined before – write it out.
Although I don’t use DCA, it seems that the outlined ovals should have some points within?
Based on what data was this graph constructed?
p16/68 - distance was lowest. Distance is not low. Shortest?
<<the distance was lowest between at site>> – ‘between at’ is not clear to me.
16/84 - fast changes – define fast.
Author Response
Comments and Suggestions for Authors
The manuscript describes in detail analyses of SOM components in 3 experimental areas, with implications on accumulation, quality and stability of organic C.
The work is very systematic, the experimental design is well planned, and the field sites are meaningfully selected. However, the ms lacks novel hypotheses or questions that would address current knowledge gaps; the results serve as a mere confirmation of previously proposed, confirmed and published hypotheses on microbial / fungal role in C stabilization in soils.
The paper is very observational and descriptive, and the outcome is very predictable - that organic C and N has been, indeed, higher under SRCs. The main contribution of the study is the field setup, with a good general relevance to the SRC areas. The ms was somehow hard to read – even papers with a lot of comparison should have a good logical flow.
Construction of some sentences was rather cumbersome - I suggest a review of a technical writer.
Despite having said all this, I trust the topic of the revised manuscript will be of interest to readers of Soil systems.
I have general and specific comments for the authors’ consideration:
General:
· L 86 - as I said, the hypothesis is very broad. What is the specific novelty of this research? Point it out.
Ø The hypothesis was revised and specified.
· Fig 2 and 3 - when comparing 6 sites, they need to be all included. Not clear how sites were selected to be shown in 2 and 3. Controls (REF) are an important part of the data presented. Specifically, Fig 2 – you compare GUL and VIP, where is the young CAH? It is listed in Table 4.
Site CAH was added in Fig. 3 (former Fig. 2). In Fig. 4 all measured treatments were presented. Information on missing treatments in Fig. 4 were added in Material and Methods.
· The whole normalization against controls was not clear. Adjusting the values of the respective analyses to controls should be more prominent.
The following information was added in material and methods: “REFs were selected within the shortest possible distance to the SRC (CAH, GUL) or former SRC (VIP) and were presumed to reveal the site-specific basic conditions under arable use in long-term absence of woody and ectomycorrhizal crops.”
Separation in soil depth sections was used as basis of comparison. The statistical analyses included evaluation on normality of the data and transformation into normality if necessary.
· Discussion was very descriptive of observed results: readers may appreciate your opinion on what could be projected, based on your results and literature, to e.g. elevated atmospheric CO2 concentration and other climate change factors. On production, turnover and allocation to ECM fungi. C sink.
The discussion was revised and recommendation for subsequent analyses were added.
Specific:
(Note: the line numbering changes throughout the ms: it starts again from line 1 at four different places: p 7, 9, 11, 14. So I will refer to page/line.)
Page 1/ line 15 – dynamics- more like composition
Revised.
1/25 - AM fungi used for the first time, write it out (it’s on 1/19).
Done.
1/28 – increased
Revised.
p2/45 – no hyphen in land use
Revised (hyphen was deleted).
48 – a cumbersome sentence
The sentence was revised.
56 – their à the
Revised.
57 - In a sentence constructed this way, it is not clear what is ‘62%’ referring to.
The sentence was revised.
56 – ‘The external mycelium …’ sentence should be placed following its intro on l. 58 – ‘The mycorrhizal …’,
The sentence was moved as recommended.
82 – ‘structure of’- is repeated.
Corrected.
86 – these hypotheses are very broad.
The hypothesis was revised and specified to the present study.
p3/128 – were taken
Corrected.
130 – by à for – streamline the sentence.
Revised.
131 - Strongest à the most strongly
Revised.
p5/223.4 – the ratio is not wider or closer. It’s higher or lower.
Corrected.
Results:
3.1. - Was the enzymatic activity of the three SRCs normalized against their REF?
The enzyme activities are based on the soil dry matter basis not on the SOM content. This is recommended by the authors of the methods used.
Fig 3 – Phospholipids: again – 2 of them are missing: ref to CAH and VIP. If you’re comparing all 6 sites, they need to be plotted.
2 treatments are missing caused by technical problems during extraction. The explanation of this aspect was added in material and method description.
VIP- FRC: ‘former’ should be previously defined. Later, fSRC is used in Fig 5. Please clarify.
FRC
The abbreviation was clarified.
p14/20 - surface-near à near-surface
Corrected.
25 - surface-near à near-surface
Corrected.
28 – disappeared – according to Fig 1, it didn’t disappear, it got reduced.
Corrected.
29 - sustainably (?) - sustained?
Corrected.
37, 38 – this belongs to Results.
The sentence was moved to Results.
Fig 4 is not mentioned in Results– it just comes in with a very little description, and no site-specificity. Just ‘soil”? What soil?
Former Fig. 4 was moved to Results (now Fig. 2) and description was added in the figure caption and in the text.
p15/57-63: the generalized priming effect correlated from literature to your experimental sites is not convincing. This paragraph would need more factual substance based on your results.
The statement was linked to the results of the present study.
p. 15 – fSRC – explain f (first time used ‘former SRC’), also unexplained in Fig 3 VIP-FRC
Explanation of the abbreviation was added.
Fig 5 – DCA has not been defined before – write it out.
DCA information was added.
Although I don’t use DCA, it seems that the outlined ovals should have some points within?
The figure was revised and the points are visible now.
Based on what data was this graph constructed?
The DCA is based on the data of Py-FIMS and PLFA-analyses of the soil of the treatments with 3 replicates.
p16/68 - distance was lowest. Distance is not low. Shortest?
Corrected.
<<the distance was lowest between at site>> – ‘between at’ is not clear to me.
Revised.
16/84 - fast changes – define fast.
Revised into “tillage-induced changes”.

Reviewer 2 Report
The manuscript of Baum and coworkers is a valuable contribution to understand soil organic matter characteristics effected by different crops, especially the effect of short rotation coppices. Generally, it is worth publishing, however, there are quite a few questions which ought to be clarified before. Let us see them in the order of their appearance.
line 28. „Increase SOM stability”?
line 39. Increased litter quantity – I guess – is a result of the postponed harvest!
lines 51-52. This is strange? Soil bulk density changes cause only relative changes in soil SOM, or?
lines 99-125. Please give the composition and coverage of the understory! This is essential in understanding the effects! The main differences were detected in the 0-5 cm soil layer, which is naturally deeply influenced by the plants. In case of SRC leaf litter effect seems to be more important, whereas in case of grasslands root deposition will be dominant. This is why the remark for lines 123-125 must be discussed!
line 117. The precipitation at VIP site is significantly higher, then on the other sites. This is not at all discussed in the manuscript! This may significantly influence e.g. general fungal, or mycorrhizal growth!
lines 123-125. This reference site totally differs from the other two. Thus this reference site will not help to „investigate the lasting effects of a former SRC”. Unfortunately, this is an important result of the investigations! On the other sites no-till (SRC) soils were compared with tilled cereal fields, while here a tilled former SRC is compared, with a no-till „grassland”!
line 126. Please give the exact dates of sampling!
Table 1. Table captions introduce the abbreviation FRC. Is this an alternate for fSRC? Please check it throughout the text!
Lines above Figure 1., and Figure 1. „The highest C-stock was measured in 15-20 cm soil depth” I do not realize this! „Under both treatments at VIP the C-stock was lowest close to the soil surface in 0-5 cm soil depth” I do not see this. What about CAH? Are these statements statistically proven?
Table 3. See the fungal signatures and compare with SOM values, and vegetation! What will be the discussion in this case?
Line 6 above Figure 2. Please discuss why flavonoids are characteristic for VIP site!
Discussion part. Please discuss all the above mentioned questions!
Author Response
Reviewer 2:
Comments and Suggestions for Authors
The manuscript of Baum and coworkers is a valuable contribution to understand soil organic matter characteristics effected by different crops, especially the effect of short rotation coppices. Generally, it is worth publishing, however, there are quite a few questions which ought to be clarified before. Let us see them in the order of their appearance.
line 28. „Increase SOM stability”?
This was clarified by revision into “Increased stability of SOM to microbial degradation…”
line 39. Increased litter quantity – I guess – is a result of the postponed harvest!
Information was added: “…an increased litter quantity caused by winter harvest in the leafless state…”
lines 51-52. This is strange? Soil bulk density changes cause only relative changes in soil SOM, or?
Increased soil bulk density (e.g. by compaction) with constant SOM concentration can increase the total SOM stock if the same soil volume per area was considering like in the present study.
lines 99-125. Please give the composition and coverage of the understory! This is essential in understanding the effects! The main differences were detected in the 0-5 cm soil layer, which is naturally deeply influenced by the plants. In case of SRC leaf litter effect seems to be more important, whereas in case of grasslands root deposition will be dominant. This is why the remark for lines 123-125 must be discussed!
Material and method: Information on the understory under SRC was added. Information on the main fine density of poplar was added. Discussion: Role of root and leave litter under SRC and grassland was discussed.
line 117. The precipitation at VIP site is significantly higher, then on the other sites. This is not at all discussed in the manuscript! This may significantly influence e.g. general fungal, or mycorrhizal growth!
Discussion of the differences in the precipitation was added.
lines 123-125. This reference site totally differs from the other two. Thus this reference site will not help to „investigate the lasting effects of a former SRC”. Unfortunately, this is an important result of the investigations! On the other sites no-till (SRC) soils were compared with tilled cereal fields, while here a tilled former SRC is compared, with a no-till „grassland”!
The no-till period under the grassland reference in VIP lasted for two years only. Unfortunately, this was the only available reference site at VIP, since the former SRC was situated between a cemetery and this site with agricultural grass for two years. We were aware of the fact that this is not optimal, but we found no good alternative in a close distance.
Information on the criteria for selection of REF sites was added in Material and Methods.
line 126. Please give the exact dates of sampling!
Information on the sampling in April 2011 was added.
Table 1. Table captions introduce the abbreviation FRC. Is this an alternate for fSRC? Please check it throughout the text!
The abbreviations were corrected.
Lines above Figure 1., and Figure 1. „The highest C-stock was measured in 15-20 cm soil depth” I do not realize this! „Under both treatments at VIP the C-stock was lowest close to the soil surface in 0-5 cm soil depth” I do not see this. What about CAH? Are these statements statistically proven?
The presentation of Fig. 1 data was corrected. A mistake was deleted. All statements are statistically proven (information on the statistically method is available in Material and Methods (section 2.7).
Table 3. See the fungal signatures and compare with SOM values, and vegetation! What will be the discussion in this case?
Information in the discussion was added.
Line 6 above Figure 2. Please discuss why flavonoids are characteristic for VIP site!
Information was added in the discussion on assumed flavonoid enrichment by fine roots and rhizodeposition under grass and cereals.
Discussion part. Please discuss all the above mentioned questions!
Information in the discussion was added.

Round 2
Reviewer 1 Report
The authors fully addressed all my comments.